# The Interplay of Hypoxia Signaling on Mitochondrial Dysfunction and Inflammation in Cardiovascular Diseases and Cancer: From Molecular Mechanisms to Therapeutic Approaches

**DOI:** 10.3390/biology11020300

**Published:** 2022-02-12

**Authors:** Esmaa Bouhamida, Giampaolo Morciano, Mariasole Perrone, Asrat E. Kahsay, Mario Della Sala, Mariusz R. Wieckowski, Francesco Fiorica, Paolo Pinton, Carlotta Giorgi, Simone Patergnani

**Affiliations:** 1Department of Medical Sciences and Laboratory for Technologies of Advanced Therapies (LTTA), University of Ferrara, 44121 Ferrara, Italy; bhmsme@unife.it (E.B.); mrcgpl@unife.it (G.M.); prrmsl@unife.it (M.P.); khsstn@unife.it (A.E.K.); mario.dellasala@unife.it (M.D.S.); paolo.pinton@unife.it (P.P.); 2Translational Research Center, Maria Cecilia Hospital GVM Care & Research, 48022 Cotignola, Italy; 3Laboratory of Mitochondrial Biology and Metabolism, Nencki Institute of Experimental Biology of the Polish Academy of Sciences, 02-093 Warsaw, Poland; m.wieckowski@nencki.edu.pl; 4Department of Radiation Oncology and Nuclear Medicine, AULSS 9 Scaligera, Ospedale Mater Salutis di Legnago, 37045 Verona, Italy; francesco.fiorica@aulss9.veneto.it

**Keywords:** hypoxia, HIF-1α, mitochondria, oxidative stress, inflammation, cardiovascular diseases, cancer, therapeutic target

## Abstract

**Simple Summary:**

The regulation of hypoxia has recently emerged as having a central impact in mitochondrial function and dysfunction in various diseases, including the major disorders threatening worldwide: cardiovascular diseases and cancer. Despite the studies in this matter, its effective role in protection and disease progression even though its direct molecular mechanism in both disorders is still to be elucidated. This review aims to cover the current knowledge about the effect of hypoxia on mitochondrial function and dysfunction, and inflammation, in cardiovascular diseases and cancer, and reports further therapeutic strategies based on the modulation of hypoxic pathways.

**Abstract:**

Cardiovascular diseases (CVDs) and cancer continue to be the primary cause of mortality worldwide and their pathomechanisms are a complex and multifactorial process. Insufficient oxygen availability (hypoxia) plays critical roles in the pathogenesis of both CVDs and cancer diseases, and hypoxia-inducible factor 1 (HIF-1), the main sensor of hypoxia, acts as a central regulator of multiple target genes in the human body. Accumulating evidence demonstrates that mitochondria are the major target of hypoxic injury, the most common source of reactive oxygen species during hypoxia and key elements for inflammation regulation during the development of both CVDs and cancer. Taken together, observations propose that hypoxia, mitochondrial abnormality, oxidative stress, inflammation in CVDs, and cancer are closely linked. Based upon these facts, this review aims to deeply discuss these intimate relationships and to summarize current significant findings corroborating the molecular mechanisms and potential therapies involved in hypoxia and mitochondrial dysfunction in CVDs and cancer.

## 1. Introduction

Cardiovascular disease (CVD) and cancer are the major issues threatening public health globally and relatively widespread with enhanced morbidity and mortality. New strategies to improve their prevention and treatment are priorities worldwide. Besides, accumulating insights reported risk factors potentially linking both disorders such as environmental factors (obesity, tobacco, sedentary lifestyle, and diet), genetics, cellular, and signaling mechanisms [1]. Furthermore, a growing number of studies demonstrate that the incidence of CVDs risk is higher in patients with cancer disorders [2] and various ancillary processes correlated to CVDs have been identified to have a role in the pathogenesis of cancer [3,4]. Therefore, understanding the mechanism overlapping in these disorders paves the way to improve and elucidate putative therapeutic targets and preventative approaches, ultimately developing emerged cardio-oncology research.

Oxygen (O_2_) delivery is an effective prerequisite to ensure the normal function of the cell and is fundamental for a wide range of physiological responses including, cell metabolism, and growth. O_2_ serves as an important for aerobic respiration that yields the primary cellular energy, the adenosine triphosphate (ATP) [5]. This process normally occurs at the powerhouse of O_2_ consumption in the cell, the mitochondria, mainly via oxidative phosphorylation (OXPHOS) and tricarboxylic acid (TCA). However, when O_2_ supply is insufficient to meet cellular energy demand, cells undergo hypoxia and are forced to use anaerobic respiration, which generates less than a tenth of the amount of aerobic respiration. Subsequently, mitochondria are severely affected by hypoxia, they sense the O_2_ levels and initiate cellular responses to hypoxia. Along with those lines, mitochondria are considered the key source of hypoxic damage in the human body [6]. Therefore, hypoxia introduces dysfunctional feedback resulting in mitochondrial damage that exacerbates oxidative stress and inflammatory signaling, correlating to mitochondrial metabolism upon hypoxia [7]. It is well known that hypoxia is a hallmark of various diseases. Indeed, at cellular levels, emerging evidence reported the pathophysiological of both CVDs and cancer disorder involve complicated and coordinated signaling pathways triggered during a decline of tissues or cells O_2_ stress (hypoxic milieu). We have selected to discuss both CVDs and cancer as major disorders characterized by the involvement of hypoxia in which most importantly has a crucial impact in promoting several processes, such as oxidative stress, inflammation, as well as cell death, and it is considered as one of the key common features with dual roles in both mentioned disorders [8].

In response to hypoxia, each cell displays numerous types of signals at the transcriptional and translational levels, consisting of the activation of a group of genes termed as hypoxic-inducible genes, which are involved in different biological processes, including cellular metabolism (lactate dehydrogenase-A (LDH-A) or pyruvate dehydrogenase kinase isoform 1 (PDK)) [9,10], angiogenesis (vascular endothelial growth factor-A (VGF-A)) [11], erythropoiesis (erythropoietin (EPO)) [12], and inflammation (inducible nitric oxide synthase (iNOS)) [13]. 

Hypoxic-inducible factor-1 (HIF-1) is a central regulator mediating the cellular response in hypoxic conditions. It is composed of a heterodimer of constitutively expressed subunits, which are the O_2_-regulated subunits HIF-α and HIF-β [14,15]. Research of Gregg Semenza’s laboratory at Johns Hopkins Medical Institutions led to HIF-1 exploration after discerning hypoxia response element (HRE), putative sequences in the 3′-flanking region of the human EPO gene. Further research found that the transcriptional activation of several regulatory genes is triggered by the binding of a particular protein to the HRE, which is induced by hypoxia. This protein was later identified as HIF-1 [16]. 

Over the last few decades, multiple pioneers of research on HIF-1α have strikingly revolutionized the comprehension of the O_2_ balance. In fact, HIF-1α has drawn much attention in many research fields, as it is outlined as the master O_2_ regulator within the cells, a hallmark transcriptional factor in the cellular response to a hypoxic environment, and a primary element for the regulation of several genes during hypoxic condition [17]. It is foreseeable that the disruption in hypoxia-related pathways contributes for several pathological states including CVDs as well as cancer, pointing out the key role of HIF-1 as a breaking point and a major cause specifically between both mentioned disorders [18]. 

This review aims to cover the actual comprehensions about the effect of hypoxia-mediating signaling pathways on mitochondrial function, and inflammation in key human diseases such as CVDs and cancer. The complexity of these impacts will be examined in the light of recent studies that shall help us to better dissecting the molecular mechanism and designing potential therapeutic approaches against both disorders.

## 2. Molecular Characteristic and Regulation of HIF-1

In mammalian cells, HIF-1 has been demonstrated to play a fundamental impact in cellular and systemic O_2_ homeostasis [19], which mediates adaptation to hypoxia through activation of a multitude of genes encoding proteins needed for improving tissue O_2_ homeostasis, energy metabolism, and efficient management of hypoxia-induced toxic stress [20]. HIF-1 is a heterodimeric trans-acting DNA-binding transcription factor that constitutionally comprises expressed subunit HIF-1β (aryl hydrocarbon receptor nuclear translocator, ARNT) and an O_2_-sensitive expressed HIF-1α subunit (or its analogues HIF-2α and HIF-3α) a master transcriptional regulator in response to hypoxia and a key modulator for the induction of genes that facilitate adaptation and survival of cells [21]. Both subunits, α and β, exhibit basic helix–loop–helix (bHLH) motifs and belong to the bHLH-Per-ARNT-Sim (PAS) homology protein family. The bHLH domain is a DNA-binding domain that can bind HREs to target specific genes [22,23].

In normal conditions, the HIF-1α subunit is hydroxylated by HIF prolyl-4-hydroxylases at proline 402 and 564 in the O_2_-dependent degradation domain (ODDD) of the α-subunits, causing its ubiquitination and proteasomal destruction via the ubiquitin-proteasome (26S) mechanism, which is able to incite constantly the proteasomal degradation. Von Hippel–Lindau (pVHL), which also acts as a tumor suppressor, binds the ubiquitin ligase complex E3, that targets HIF-1α subunit destruction in O_2_-dependent degradation domain. Because of this, during normoxia HIF-1α protein has a short half-life [24,25]. In contrast, under hypoxia, the repression of O_2_-dependent propyl-hydroxylase-1, -2, and -3 enzyme activity (PHD1, -2, -3) results in suppression of HIF-1α hydroxylation. HIF-1α protein is then stabilized, accumulates in the cytosol, and transferred into the nucleus, where it creates a heterodimer complex with HIF-1β and binds to HREs with a consensus sequence (5′-RCGTG-3′) in promoter or enhancer of target genes to activate a concerted transcriptional response (Figure 1). The nuclear translocation of HIF-1α is not enough to stimulate the target genes transcription [23]. The HIF-1α/HIF-1β (HIF-1) transcription factor recruits various cofactors that are fundamental for full transcription activity, including CREB-binding protein (CBP)/p300 and transcription intermediary factor 2 steroid-receptor activator that ultimately binds to CTAD domain. Another cofactor recognized is the M2 isoform of pyruvate kinase (PKM2), which enhances the binding of the complex HIF-1 to HRE [26].

## 3. Role of Hypoxia Signaling and Mitochondria in Cardiovascular Diseases (CVDs) and Cancer 

### 3.1. Hypoxia Signaling and Mitochondria in CVDs

Hypoxia stimulates multiple processes to adapt to insufficient levels of O_2_ in the environment. Therefore, it has mostly negative consequences for cardiovascular functions, ultimately manifesting in pathology. HIF-1α is an O_2_-sensitive transcription factor that regulates adaptive metabolic responses to hypoxia and elicits a crucial impact in various CVDs, such as ischemic heart disease (IHD) and heart failure (HF) [27]. Recent insights demonstrated the effect of HIF-1α signaling in the progression of heart disorders [28,29], or its cardioprotective role after I/R in animal model deficient of PHD3 and HL-1 cardiomyocytes [30,31] (Figure 2). Furthermore, HIF-1α overexpression caused mitigation of ischemia/reperfusion (I/R)-enhanced cardiomyocytes loss, suggesting that HIF-1α may drastically influence cardiomyocytes endurance [32]. Recently, studies documented that the HIF signaling pathway not only stimulates disease progression but has a cardioprotective effect as well as the potential for cell recovery from cellular stress in various disorders. This seems to be linked to the duration of hypoxic exposure as well as the stabilization of HIF-1. In fact, chronic exposure to hypoxia is found to increase ischemic ventricular arrhythmias and further cell death [33]. While intermittent exposure to hypoxia reduces arrhythmia during I/R, as also stimulates protective effects against myocardial infarction in rodents [34,35,36,37]. The prolonged HIF-1α upregulation was shown to promote dilated cardiomyopathy in transgenic mice with PHD2 depletion [38]. HIF-1α can also influence mitochondrial function and alleviate the severity of ischemic heart. HIF-1α can also influence mitochondrial function and alleviate the severity of ischemic heart. HIF-1α can upregulate mitophagy, mitochondrial autophagy, in cardiac cells through HIF-1α/(BNIP3) BCL2 and adenovirus E1B 19-kD-interacting protein 3 pathways, thereby stimulating their survival following myocardial ischemia-reperfusion. This is only applied to the role of HIF-1α-mediated mitophagy at an early phase of ischemia, which may result in cardiac protection, while prolonged autophagy may activate cell death in H9C2 cardiomyoblasts and Sprague Dawley rat models [39]. In contrast, other studies revealed that HIF-1 activation enhances BNIP3 expression, resulting in (H9C2) cardiomyocyte death, which is a hallmark of ischemia and HF [40,41] (Table 1). Major interplays have been identified between hypoxia-mediated mitochondrial function and mitophagy in cardiomyocytes. In brief, upregulation of the inner mitochondrial membrane protein (IMM), Optic atrophy 1 (Opa1) stimulates mitophagy and mitochondrial function in response to hypoxia in mouse cardiomyocytes [42]. FUN14 Domain Containing 1 (FUNDC1) is an OMM protein that accumulates on the mitochondrial associated membranes (MAMs). Several recent studies reported its effective role to mediate mitophagy during ischemic conditions in cardiomyocytes, and it thus conveys cardioprotection [43]. Although the cardioprotective effect of mitophagy in the ischemic heart is widely demonstrated, during the reperfusion stage, mitophagy has a defective impact on cardiac function, and this may be due to the repression of FUNDC1-dependent mitophagy and necrosis upregulation [44]. The contribution of FUNDC1 in response to hypoxia may provide new insight in favor of therapeutic target approaches in CVDs, and further research focused on the FUNDC1-HIF-1 axis may be beneficial. Furthermore, a novel protein, WD Repeat Domain 26 (WDR26), has been detected to localize into the mitochondria, promoting mitophagy in H9C2 cells during hypoxia, suggesting its pivotal effect in hypoxia-enhanced mitophagy [45]. In another line of evidence, HIF-1α accumulation directs mitophagy and promotes the differentiation of H9C2 cells [46]. Therefore, HIF-1 modulates hundreds of genes in diverse biological pathways, and most of them influence mitochondrial function. HIF-1 activates mitochondria-specific genes crucial to a metabolic shift away from OXPHOS to glycolysis, including LDH-A, phosphoglycerate kinase-1 (PGK1), to adapt to hypoxic stress [47]. HIF-1 elevates glycolysis by upregulating glycolysis enzyme production, increasing glucose transporters, and repressing the mitochondrial energy metabolism [18]. Moreover, HIF-1α promotes PDK-1 activation, which phosphorylates and inhibits pyruvate dehydrogenase (PDH), from converting pyruvate to acetyl CoA to fuel the mitochondrial TCA cycle and preventing the formation of iron-sulfur (Fe/S) clusters, thereby attenuating complex I activity [48]. It also inhibits the expression of mitochondrial encoded subunits in OXPHOS complexes by blocking the nuclear–mitochondrial interaction [49]. Recent pieces of evidence have also reported HIF-1α in improving mitochondrial function, reducing cellular oxidative stress, and stimulating the cardioprotection [39]. 

Furthermore, HIF-1α improves mitochondrial respiratory function by triggering various cardioprotective signaling pathways, including the phosphoinositide-3-kinase/Akt (PI3K/AKT) and Janus kinase (JAK) 2/signal transducer and activator of transcription (STAT) 3, to protect the heart during I/R injury [50]. Indeed, Nanayakkara and colleagues reported the transcriptional role of HIF-1α during hypoxia in regulating frataxin expression levels, a highly conserved nuclear-encoded mitochondrial protein, expressed in tissues such as the heart, neurons, and liver with a high metabolic rate [51], which served as a cardioprotective element against ischemic injury. Ultimately, enhanced frataxin levels can alleviate mitochondrial iron overload, thereby preserving mitochondrial membrane integrity and the cardiomyocyte’s viability [52]. HIF-1α stabilization permits cells and tissues to adapt to the hypoxic response in I/R, thus protecting cardiomyocytes against IHD and improving patient prognosis [20]. On the other hand, several lines of evidence outlined the impact of HIF-1α localization dependent on the mitochondrial function regulation, which has been thought to be contingent on its trafficking to the nucleus. Nevertheless, studies documented that HIF-1α not only localizes to the nucleus after exposure to hypoxia or preconditioning but surprisingly it localizes also to the mitochondria [53,54]. For instance, Mylonis and collaborators reported that HIF-1α at the outer mitochondrial membrane (OMM) attenuates hypoxia-induced apoptosis [55]. In alignment with this regard, during the elevation of oxidative stress, HIF-1α translocates to the mitochondria to reduce mitochondrial reactive oxygen species (mtROS) in the human umbilical vein endothelial (HUVEC) during hypoxia [56].

### 3.2. Hypoxia Signaling and Mitochondria in Cancer

Multicellular organisms have adopted several mechanisms to rapidly adjust to hypoxia, prolonging survival in the absence of adequate resource [61]. A common observation of most tumors is an insufficient amount of O_2_, the severity of which varies between tumor types. In proliferating and expanding tumor tissue, the adaptation of tumor cells to exhausted oxygen supply is mainly mediated by HIF-1. Such metabolic adjustment is pivotal for cancer cell survival and proliferation in response to environmental stimuli [62].

In this section, we will discuss how hypoxia affects the mitochondrial function through TCA citric acid cycle, electron transport chain, and its dual role of hypoxia in ROS production and mitigation and finally briefly consider hypoxia-induced mitochondrial distribution and morphology in cancer. 

#### 3.2.1. Hypoxia-Induced Modulation of Krebs Cycle and Oxidative Respiration

TCA cycle represents the metabolic engine within cells. It is found to be inhibited by HIF-1α through the induction of the enzyme PDK1. PDK1 inactivates the TCA cycle enzymes by phosphorylating PDH, which converts pyruvate to acetyl-CoA. PDK1 overexpression in HIF-1α-silenced cells led to an increased ATP production, reduced ROS generation and prevented the hypoxia-mediated apoptosis [9]. Indeed, the accumulation of 2-hydroxyglutarate (2-HG), an essential epigenetic regulator in cancer cells, has been documented to enhance the stabilization of HIF-1α [63]. Importantly, HIF-1α was found to act as a metabolic switch from glycolysis to OXPHOS for regulatory T cells glioblastoma. Specific ablation of HIF-1α in regulatory T cells resulted in enhanced pyruvate import into mitochondria [64]. Taken together, TCA cycle metabolites resulted to be affected by hypoxia to limit substrate availability for phosphorylation and epigenetic modifications to change cell function and fate.

In addition, the effect of HIF-1α-mediated metabolic reprograming also alters OXPHOS respiration [65]. Evidently, MCF-7 carcinomas cells exposed to 24h hypoxia showed reduced OXPHOS flux and decrease 2-OG dehydrogenase as well as glutaminase activities, without functional alteration of respiratory complexes I and IV [66] (Table 2). Other possible ways by which hypoxic cells reduce oxidative metabolism involve small RNAs. HIFs increased transcription of genes encoding microRNAs (miRs), small RNAs that link to mRNAs in sequential mode to either suppress their translation or promote their degeneration. Among the list of targets of miR-210, there are the iron-sulfur cluster enzyme (ISCU) genes, required for mitochondrial complex I function, which are particularly found downregulated under hypoxia [67]. Moreover, a bioinformatics survey and PCR real-time experiments demonstrated the involvement of both nuclear respiratory factors (NRF-1) and HIF-1α in modulating voltage-dependent anion-selective channel 1 (VDAC1) promoter during nutrients deprivation or hypoxia [68]. Taken together, cancer cells activate a metabolic switch that involve HIF-1α and impaired Krebs cycle and OXPHOS function to sustain glycolysis metabolism.

#### 3.2.2. Hypoxia-Induced Mitochondrial ROS Production and Suppression 

ROS are actively generated in the form of superoxide and hydroxyl free radicals as a by-product of OXPHOS and neutralized by antioxidant mechanism to ensure a proper cellular function [79]. Indeed, in breast cancer cell line (MDA-MB-231), the inhibition of HIF-1α expression using cardamonin repressed the mechanistic target of rapamycin (MTOR) pathway causing increased OXPHOS activity and enhanced ROS, which finally led to apoptosis [71]. Similarly, colorectal cancer cells under hypoxic conditions are characterized by activation of OMA1-OPA1 axis, which in turn increase the mtROS generation to stabilize HIF-1α, thereby promoting the glycolytic metabolism [75]. There is also evidence that mitigating ROS-mediated damage in hypoxia may promote the Warburg effect, an altered metabolism favoring enhanced uptake and use of glucose. Warburg effect has been also involved during the pathogenesis of cancer cachexia (CC), which is a complex pathological condition with metabolism dysregulation, affecting primarily the skeletal muscle [80]. The impaired mitochondrial homeostasis and metabolism has also been documented in various models of CC. Additionally, it has been suggested that HIF-1α may have an impact on the metabolic alteration in cancer cachexia (CC) [77]. Consistently, suppressing HIF-1α expression by Rhein and Emodin compounds mitigates cancer cell proliferation and CC in a dose- and time-dependent mode [81,82]. 

This indicates that the alleviation of ROS may promote the antioxidant system. Particularly, stimulation of AMP-activated protein kinase (AMPK) by mitochondrial ROS led to peroxisome proliferator-activated receptor gamma coactivator 1-alpha (PGC-1α)-dependent antioxidant response. Mechanically, AMPK-PGC-1α induced regulation of mitochondrial ROS mediates the HIF-1α stabilization [83]. Interestingly, opposite to earlier experimental observations, downregulation of superoxide dismutases (SOD2) expression has also been observed under hypoxia in a HIF-1 dependent manner in renal carcinoma cells [84]. Similarly, HIF-1 increased glutathione levels by upregulating glutamate cysteine ligase (GCLM) in breast cancer cells (MDA-MB-231 and SUM-149 cells) [72]. 

#### 3.2.3. Hypoxia-Induced Mitochondrial Distribution and Dynamics

The mitochondrial distribution and dynamics are permitted by a correct balance between two opposite processes, that are the mitochondrial biogenesis and the mitochondrial fission. In addition to this, mitochondria removal mechanisms which permit the elimination of damaged and excessed mitochondria guarantees the preservation of a healthy mitochondrial pool. Among the diverse mechanisms regulating this mitochondrial turnover, the selective form of autophagy pathway, known as mitophagy [85], represents the primary line of intervention. Dysfunctions in such mitochondrial-related pathways may be cause and consequence of several disorders [86,87,88,89] and can be also related to a hypoxic condition. For example, it has been observed that cancer cells activate mitochondrial fusion to enlarge their mitochondrial population to evade from cell death stimuli. Consistently, inhibition of the mitochondrial fusion protein mitofusin-1 (MFN-1) reestablish a tubular network [76]. Other factors appear to influence the hypoxia-mediated fission of mitochondrial membranes. This is the case of the mitochondrial scaffolding protein A-kinase anchor protein 1 (AKAP121). During hypoxia, AKAP121 regulates mitochondrial dynamics through inhibition of phosphorylation of the main regulators of mitochondrial fusion events: the dynamin-related protein 1 (DRP1) and the fission 1 protein (FIS1). Interestingly, this work also accounted for the ubiquitin ligase SIAH2 a crucial role in regulation of hypoxia-mediated mitochondrial fission. Indeed, it is well documented that AKAP121 is a SIAH2 substrate and, of consequence, the availability of AKAP121 as well as the associated mitochondrial dynamics under hypoxia are highly dependent of SIAH2. Consistently, SIAH2 knockout (KO) cells displayed higher AKAP121 levels [90]. In addition, the mitochondrial biogenesis has important roles in cancer cells to adapt hypoxia. Microarray studies show that reconstitution of pVHL in renal carcinoma led to elevated mitochondrial mass, mitochondrial complexes activity and O_2_ consumption rates. Mechanistic insights show MAX-interactor 1 (MXI1) expression reduces C-MYC-dependent expression of PGC-1α, which in turn inactivate mitochondrial biogenesis [69]. Similarly, ablation of high mobility group box 1 (HMGB1) protein in hepatocellular carcinoma subjected to hypoxia promoted mitochondrial biogenesis and reduced ATP. Mechanistic experiments suggest that the binding of HMGB1 to Toll-like receptor-9 in cytoplasm stimulate p38 which in turn led to phosphorylation of PGC-1α, which resulted in subsequent increase mitochondrial biogenesis [70]. There are several reasons given why tumor hypoxia maintain increased mitochondrial biogenesis. Thanks to this condition cancer cells result protected from cell damage induced by ROS, thereby providing a survival mechanism. Another benefit is to enhance their migratory and invasive properties to other zones with less hypoxia [91]. 

As reported above, central to keeping healthy mitochondrial population and guarantee and efficient turnover of the mitochondrial population is mitophagy [92]. The exact role of mitophagy in cancer remains unresolved, as it has divergent roles, either blocking or stimulating the cancer progression [93]. Interestingly, growing consensus documented an intimate relationship between mitophagy and hypoxia in regulate the tumor growth. Indeed, the mitophagy proteins BNIP3L/NIX, BNIP3, and FUNDC1 were found upregulated in response to hypoxia in cancer [94,95,96]. Furthermore, BNIP3 and NIX are target genes of HIF-1α and their upregulation stimulates mitophagy under hypoxia and inhibits cancer progression in MMTV-PyMT model under hypoxia through activated HIF [73,74]. In addition to this, the protein E2F3d, recently stated to localize to the OMM, revealed to augment the process of mitophagy during hypoxic exposure in cancer cells [97].

During hypoxia, HIF-1α regulates glucose, glutamine metabolisms, and lipid through various target genes, by which it enhances glycolysis by upregulating glucose transporters to induce the glycolysis flux and targeting PDK-1/3 to inhibit pyruvate changeover to acetyl CoA and stimulating LDH-A to turn the pyruvate to lactate. HIF-1α upregulates glutamine utilization and activates fatty acids for rapid cell growth and division. Moreover, when O_2_ supply is limited, tumor cells are protected against excessive ROS and subsequently they resist to cell death.

## 4. Mitochondrial Dysfunction and Inflammation in CVDs and Cancer

### 4.1. Oxidative Stress and Mitochondria

Under physiologic conditions, other organelles in addition to mitochondria have the capacity to produce ROS such as peroxisome and endoplasmic reticulum (ER) [98]. Despite this, several lines of evidence ascribe to mtROS a predominant role in CVDs as well as in cancer (as reviewed in [99,100,101]). In the mitochondrion, ROS are produced by ETC in the IMM at the level of complex I and III during oxidative metabolism and cellular response to cytokines and bacterial invasion. Moreover, several enzymes here located or translocated following stimuli are considered additional sources of mtROS [102], and they include several flavoproteins (acyl-CoA dehydrogenase, glycerol α-phosphate dehydrogenase, α-ketoglutarate dehydrogenase) [103,104,105], the monoamine oxidases (MAOs) on OMM that produces ROS in the catabolism of neurotransmitters [106], aconitase [107], the ROS-generating NADPH oxidases (NOXs) [108], and p66shc in the intermembrane space (IMS) which oxidizes cytochrome c [109,110]. All these enzymes share the release of discrete amounts of superoxide (O_2_^−^) and hydrogen peroxide (H_2_O_2_) which have been linked to a consequent mitochondrial swelling and apoptosis [109]. Moreover, the O_2_^−^ produced in these steps is able to react also with nitric oxide (NO) giving rise to reactive nitrogen species (RNS) production and consequent nitrosative stress, which involves mitochondria because affects their enzymatic activity, modifies mitochondrial respiration and increases mitochondria-mediated cell death [111]. 

Uncontrolled ROS and RNS production, generate oxidative stress with consequent mtDNA damage and oxidation of proteins and lipids of membranes. In the first case, ROS and RNS directly react with pyrimidine and purine bases, especially in the D-loop region [112]; here, this damage led to a significant decrease in mtDNA copies and of ETC function [113]. Otherwise, peroxidation of mitochondrial lipids and proteins alters the mitochondrial membrane potential (MMP), their energy production and triggers the opening of mPTP [105,109,110]. Subsequently, mitochondria result impaired, and several cell death pathways are activated.

mtROS scavenging systems exist and they represent the first line of defense against toxic ROS levels. Superoxide dismutase (SOD) is responsible for the conversion of O_2-_ in H_2_O_2_. SOD exists in three isoforms and, among them, SOD2 has a mitochondrial matrix localization, while SOD1 (Cu/Zn SOD) may be partially found in the intermembrane space. A second line of defense is covered by catalase, which splits in O_2_ and water [114]. The action of this enzyme is flanked by the glutathione peroxidase/reductase (GSH-PX) and the peroxiredoxin/thioredoxin (PRX/Trx) systems which taking advantage from the reduced forms of GSH and PRX, they convert H_2_O_2_ in water. PRX3 and PRX5, as well as Trx2 are localized to mitochondria.

### 4.2. Inflammation and Mitochondria

It is currently recognized that ROS induces inflammation [115,116] and, in turn, inflammation further sustains mitochondrial dysfunction [117]. Mitochondria are organelles involved in the inflammatory process not only to produce ROS, but because in particular conditions (i.e., the opening of the mPTP or the permeabilization of the OMM) [118,119], they release into the cytosol several factors named as DAMPs (danger-associated molecular patterns). To date, many substances are recognized as mtDAMPs, and they include mtDNA [120], ATP [121], TFAM [122], NFP [123], succinate [124], and cardiolipin [125]. These molecules, once in the cytosol and sometimes in the extracellular milieu, are recognized by other molecules (adaptors or receptors) to trigger inflammation. There are different pools of receptors. Those recognizing mtDAMPs, which link mitochondria to inflammation, are the cytoplasmic nucleotide-binding domain leucine-rich repeat-containing receptors (NLRs). The most common and well-defined is the NLR and pyrin containing protein 3 (NLRP3). It exists as a multiprotein complex composed of the scaffold protein NLRP3. Normally localized in the cytosol, NLRP3 moves to the mitochondria and MAMs upon activation, and it regulates the innate immunity by recruiting the apoptosis-associated speck-like protein (ASC; an adaptor) and procaspase-1, which becomes matured in caspase-1 and is responsible for the activation of IL-1β and IL-18 [115]. The contribution of mitochondria in NLRP3-mediated inflammation is demonstrated by their recruitment in the cytoplasm of macrophages in a process mediated by microtubules [126] and the observation of NLRP3 relocalization to mitochondria following stressors. Indeed, under resting conditions, NLRP3 is normally localized at the ER, but after stimulation with mtDAMPs, NLRP3 clusters at mitochondria and at MAMs [124]. Thus, inflammation and mitochondria are directly related and together with ROS contribute to both CVDs and cancer. 

### 4.3. Inflammation, Oxidative Stress, and Mitochondrial Dysfunction following Hypoxia in CVDs

Inflammation and oxidative-nitrosative stress are two main contributing causes in the onset and progression of CVDs [127]. In addition being crucial in the excitation-contraction coupling of the heart [128], mitochondria are at the crossroad between both routes, as these are often accompanied by an alteration of mitochondrial function. Inflammation has a key role in CVDs, as an example, atherosclerosis, plaque and vessel calcification, and post-ischemic pathologies. Atherosclerosis born from cholesterol, fat, and other substances deposition on the inner side of vessels. These deposits build up into the vessel in concomitance of endothelial dysfunction, the starting artery lesion, to lead to the plaque formation. Direct evidence of the regulation of lipid metabolism by hypoxia and HIF-a still lack in CVDs. However, it has been suggested that hypoxia-induced HIF-1α accumulation upregulates the expression of its target genes and affects the lipid metabolism of hypoxic macrophages in atherosclerosis, thereby providing evidence of a possible atherogenic role for hypoxia [129]. Lipids can be also modified by further oxidation from newly generated mtROS, and adhesion molecules (ICAM and VCAM) are activated to trigger the binding of inflammatory cells, such as monocytes. Often, cholesterol deposition represents the second hit in NLRP3 assembly for the production of interleukins and inflammation progress [130]. Inflammation is also responsible for the calcification of the plaque; indeed, activated macrophages promote the apoptosis of interstitial cells by releasing proinflammatory cytokines and promoting the release of vesicles reached in calcium (Ca^2+^) and phosphates [131,132]. Similarly, an increase of aged mitochondria, reduced biogenesis, impaired mitophagy, and dysregulated cytosolic Ca^2+^ homeostasis are all associated with mitochondrial dysfunction [76,129,130]. 

On the other hand, mitochondrial oxidative stress has been found closely related to several risk factors for CVDs, and from studies in vitro and in vivo it has been found to be one of the main mediators of apoptosis in cardiomyocytes and endothelial dysfunctions [101,133]. The contribution of each of the above-mentioned ROS-producing enzymes in CVDs has been analyzed overtime in experimental settings including genetic manipulation of animal models. As an example, p66SHC is considered an important modulator of mitochondria-mediated cell death as it regulates MMP and ROS when localized to the mitochondria [134]. Accordingly, animals lacking p66SHC display reduced ROS and decreased cell death in a model of hypertension and HF [135]. Elevated levels of angiotensin II upregulate redox-sensitive pathways causing mtROS overproduction, cardiomyocyte damage, and finally hypertension and HF. The p66SHC^−/−^ phenotype benefits in cardiac function are not only against the previously mentioned pathologies but also in a model of diabetic cardiomyopathy [136]. Likewise, the inhibition or deletion of isoform A of MAOs in murine models decreases mtROS production, preserves the heart against pressure overload, caspase 3 activation, and fetal gene reprogramming [137]. In addition, MAO-B targeting in a mouse model of induced pressure overload minimizes the worsening of the cardiac function and reduces death by preserving MMP and mitochondrial bioenergetics [138]. NOX isoforms, NOX4, constitute the main source of oxidative stress during HF [139]. In support of this, its levels are significantly elevated following chronic pressure overload. Consequently, oxidative stress, mitochondrial dysfunctions, and apoptosis are activated. Mice lacking NOX4 have lower left ventricular (LV) dysfunction with reduced oxidative stress and cardiomyocyte apoptosis [140].

The dysregulation of the antioxidant systems is the second aspect that demonstrates how mitochondrial oxidative stress is deleterious in heart disease. Studies from SOD2^−/−^ mouse model reported either premature lethality [141] or deaths before 4 months due to HF. Indeed, the absence of this enzyme triggers excessive oxidative stress in mitochondria with the overproduction of 4-hydroxynonenal (4-HNE), produced from lipid peroxidation and a player in mitochondrial dysfunction, by acting via the severe impairment of the ETC [142]. 

The deriving phenotype consists of abnormalities in mitochondrial structure, decreased ejection fraction, dilated, and dysfunctional LV, all major phenotypes of dilated cardiomyopathy [142]. Deleterious effects are not limited to the complete deletion of SOD2. In a study in humans, the single Ala16Val SOD2 polymorphism has been investigated as associated with coronary heart disease (CHD) risk, in concomitance with decreased SOD2 activity and increased mtROS production [143]. In addition, endothelial dysfunction is increased when SOD2 is absent [144]. In the end, the inappropriate activity of GSH-PX and PRX/Trx systems enhance oxidative stress, which is the main cause of left ventricle (LV) contractile dysfunction and abnormalities in LV mass, under the action of angiotensin II [145]. If we consider these genotypes following I/R, they produce larger infarct size, alteration of cardiac contractility, cell death, and adverse events such as HF and dilated cardiomyopathies [146,147].

Overall, in understanding the importance of all these compensatory mechanisms in humans, an extensive meta-analysis on thousands of papers published and as many patients analyzed, reported an inverse and significant correlation between antioxidant systems activity (and circulating levels of enzymes involved in) and the most common risk factors of CHD [148].

Hypoxia is a bio-clinical condition in which either portion or whole tissues are subjected to a significant imbalance between O_2_ consumption and blood perfusion [149]. It is often associated with nutrient deprivation. In normal conditions, the cardiovascular system is appointed to guarantee these functions in the human body. However, coronary artery disease (CAD) (i.e., atherosclerosis) can arise with aging and become responsible for a progressively decrease in the blood perfusion. As a consequence, CAD may result in ischemic heart disease (IHD), such as myocardial infarction (MI) and peripheral arterial diseases (PAD), in which hypoxia plays a predominant role. In addition, being that hypoxia is a condition that induces changes in the cardiovascular system, its chronic presence can also lead to hypertension and HF [150]. Thus, within certain limits and with different degrees, all CVDs mentioned in the previous paragraphs are linked to hypoxia. In normoxia, HIF-1α binds the pVHL for degradation by the proteosome; under hypoxic states, HIF-1α accumulates and translocates mainly to the nucleus where it dimerizes with beta subunit. Being a transcription factor, in the nucleus it triggers the expression of several genes involved in either adaptation or maladaptation to hypoxia [151]. In this context, whether cells and tissues find a means to adapt in the absence of O_2_ levels, the reperfusion phase (if any) emphasizes the hypoxia-induced damages even more. Examples of adaptations induced by hypoxia are the HIF-1α and VEGF-mediated angiogenesis in post-MI hearts to ensure a compensated O_2_ delivery in the infarcted tissue [57], and the downregulation of mitochondrial O_2_ consumption through the activity of the PDK1, which is the enzyme that limits the use of pyruvate at mitochondrial level [48]. Here, the O_2_ available for mitochondria increases and consequently cell death is reduced. Again, HIF-1α overexpression is important for mitochondrial metabolic adaptation to a persistent state of hypoxia in the heart, which distinguishes patients with cyanotic congenital heart disease (CCHD) [152]. Moreover, HIF-1α KO mice in which pressure overload is induced, immediately present severe traits of hypertrophy when compared to their wild-type littermates [60]. Note that it is no coincidence if hypoxic conditioning is considered as a non-pharmacological therapeutic intervention in the adaptation of the body against severe O_2_ deprivation episodes [153]. However, chronic or sustained hypoxia states and all associated prosurvival pathways are better associated to cancer. In full-blown IHD, mechanical methods and pharmacological treatments are usually suddenly applied to solve the ischemic phase, also known as the cause of the necrotic core of the lesion. One of them is the reperfusion of the occluded vessels. 

Mitochondria are the major consumers of O_2_, and as consequence, they suffer from its absence. Whether in the normoxic condition the cell respire in the aerobic mode and produce big amount of ATP, under hypoxia ETC is inhibited in favor of glycolysis which is upregulated with consequent acidification of the environment, sodium and Ca^2+^ overload and breakdown of ATP production [154,155]. In the reperfusion phase, the reactivation of the ETC and the restoration of the MMP prompt further Ca^2+^ intake and a burst of mtROS production. These two events increase the susceptibility of mPTP to open [156]. Ca^2+^ enters the organelles by the mitochondrial calcium uniporter (MCU) complex, a highly selective channel in the IMM, ensuring Ca^2+^ uptake [157]. Indeed, cardiac-specific knockdown of MCU increases resistance against mPTP opening and reduces infarct size following ischemia [158,159]. On the other hand, most mtROS derive from an accurate and selective metabolic process which has been investigated by Chouchani and co-workers in 2014. By analyzing several ischemic and re-perfused tissues (i.e., brain, kidney, heart), they identified a mitochondrial metabolite that increased in concentration 19-fold during hypoxia: succinate. It accumulated proportionally to the time of the ischemic phase, and it was rapidly metabolized at reperfusion time, just when ROS increased. In this work, they supported the hypothesis that succinate comes from the reduction of fumarate by the reversal mode of succinate dehydrogenase (SDH) action. Indeed, fumarate also increases in ischemia through the activation of malate/aspartate shuttle and purine nucleotide cycle [160]. Succinate at reperfusion would produce mtROS by acting on complex III and inducing the reverse mode of complex I; in addition, it behaves like a DAMP to sustain inflammation. Thus, both Ca^2+^ and ROS constitute a second hit for the mPTP opening, following the ischemic priming phase. mPTP gives rise to and is accompanied by most mitochondrial dysfunctions that occur following hypoxia, including the mitochondrial permeability transition, the collapse of the MMP, and loss of cristae morphology and several proteins into the cytosol that trigger cell death and/or inflammation. In turn, mPTP opening stimulates mtROS production and all together facilitate the disassembly of super complexes of the ETC (i.e., complexes I+III+IV) further increasing ROS generation [161]. There is evidence of a strong upregulation of NLRP3-dependent inflammation following hypoxia both in cardiac fibroblasts and cardiomyocytes, which seems to have different but complementary roles. In cardiac fibroblasts, NLRP3 upregulation would sustain inflammation by IL−1β release and contribute to cardiac remodeling; in cardiomyocytes it induces pyroptosis by the solely activation of mature caspase 1 [162,163].

### 4.4. Hypoxia-Mediating Signaling Pathways and Cell Death in CVDs

Programmed cell death such as apoptosis, necrosis, and ferroptosis occur in cardiomyocytes and they are considered as central features in the pathogenesis of CVDs. In addition to the studies mentioned above on hypoxia-related oxidative stress and inflammation, several lines of evidence reported the involvement of hypoxia in mediating programmed cell death “apoptosis” in cardiomyocytes. Increased expression of HIF-1α upregulated the apoptotic effect in cardiac (H9C2) and renal ischemia (HK2) [164]. Furthermore, prolonged HIF-1α expression upregulates the activity of p53 tumor suppressor, and consequently, stimulates cardiomyocytes apoptosis following MI [58]. HIF-1α–mediated apoptosis may also have a central effect on HF. Indeed, as documented in some lines of evidence, a high level of HIF-1α expression was identified in the late phase of HF [28,165]. Notably, the induction of apoptosis in CVDs upon the HIF-1α axis is controversial, in which HIF-1α exerts a dual role in the heart during hypoxic exposure. In one hand, it acts as a cardioprotective through its stimulation to several genes and pathways to adjust to hypoxia. On the other, it stimulates cardiomyocyte’s damage through the activation of various cell death pathways [166]. Hypoxia-induced HIF-1α accumulation blocks the apoptotic process by stimulating angiogenesis and declining fibrosis [167,168]. Along with that, another mechanism in which HIF-1α prevents cardiomyocytes loss is by giving rise to iNOS and cardiotrophin-1 (CT-1) expression [169]. HIF-1α exhibits a protective effect mostly during I/R injury [170] (Table 1). Similarly, during ischemic postconditioning, the increased level of HIF-1α expression improves MI and reduces cardiac damage [171,172]. It is not surprising that HIF-1α overexpression could alleviate apoptosis through the NFκB pathway under hypoxia in MI [173]. Consistently, other lines of evidence confirmed the cardioprotective effect of HIF-1 predomination against apoptosis via the impede of Bax [174]. Besides apoptosis, necroptosis and ferroptosis are nonapoptotic types of programmed cell death that occur in numerous disorders particularly in HF, I/R, and MI [174]. Hypoxia mediates various events including, oxidative stress, and mitochondrial deregulation thereby may stimulate cardiomyocyte necroptosis. Indeed, this process has been demonstrated to contribute to MI pathogenesis [175] and in the alteration of cardiac activity during chronic ischemia [176]. Necroptosis also has a prominent effect in cardiomyocytes loss during acute viral myocarditis, I/R injury, and atherosclerosis through its ligand death receptors including receptor-interacting protein 1, 3 (RIP1, 3) and mixed lineage kinase domain-like (MLKL) [177,178]. Recently, Karshovska and associates documented that a high level of HIF-1α regulation impairs the mitochondrial bioenergetic and elevates macrophage necroptosis in the atherosclerosis [179]. Another type of regulated cell death is ferroptosis, which consists of intracellular iron dependence. Its fundamental role has been effectively stated recently in various CVDs, including cardiomyopathy, HF, MI, and myocardial I/R [59,180,181,182]. The iron chelator deferoxamine (DFO) is an iron chelator able to mimic hypoxia and to upregulate HIF-1 that can impede the process of ferroptosis in cardiac cells and eventually alleviates cardiac injury [183,184]. DFO is also able to reduce MI and to improve cardiac activity in myocardial I/R models [185,186]. Strikingly, high levels of ferroptosis have been also observed during the reperfusion stage [187]. Several other lines of evidence document the role of programmed cell death in the modulation of inflammation, which may result in cardiomyocyte ferroptosis [188]. These current findings suggest that blocking ferroptosis may prevent cardiomyocytes death. However, the processes by which hypoxia modulates cardiomyocytes death remain to be thoroughly investigated. Further specific impact of hypoxia precisely HIF-1 on ferroptosis in the heart still to be well covered.

### 4.5. Inflammation, Oxidative Stress, and Mitochondrial Dysfunction following Hypoxia in Cancer Disease

Accumulating evidence demonstrates that inflammation is involved in all stages of tumorigenesis, including limitless replication, invasion and metastasis, apoptosis evasion, DNA damage, and angiogenesis. It remains unclear which molecular mechanism interconnects all these pathways. Recent scientific evidence suggest that the mitochondrial compartment may be the central platform for the regulation of the inflammatory response, which occurs during cancer development and growth. The ATP generated by mitochondria throughout OXPHOS is fundamental for the proliferation and differentiation of T cells, which are one of the main components of antitumor immunity [189]. However, different metabolic demands are required for each phenotypic stage of T cells. Indeed, upon their activation, T cells quickly shift to glycolysis, which guarantees a rapid energetic availability to support their growth and the production of biosynthetic factors. To boost the glycolytic pathway, the molecular axis composed of PI3K, AKT, and MTOR is the main executor. On one hand, PI3K-AKT-MTOR activates the avian myelocytomatosis virus oncogene cellular homolog (c-Myc) to increase the activity of key enzyme of glycolysis (such as the glucose transporters (GLUT)) [190] and of glutamine transporters (including glutaminase1) [190,191], and on the other, the PI3K-AKT-MTOR axis works to upregulate HIF-1α, which, at the same time, can inhibit the TCA cycle and upregulate the expression of glycolytic enzymes. In addition to this, HIF-1α-induced glycolysis is fundamental to control the subpopulation of T cells. Indeed, it regulates the differentiation of Th17 [192] and of CD8+ T cells [193]. By contrast, the transition of T cells from effector to memory states mainly relies on fatty acid oxidation (FAO). Recent studies demonstrated that during this transition phase, a small amount of T cells enforces FAO by reducing the MTOR pathway and activating the AMPK pathway [194]. In addition, AMPK and FAO support the development of CD8+ T cells. Indeed, by using the AMPK activator such as metformin, the amount of memory cells as well as the lipid oxidation were increased [195]. Consistent with this, in AMPK-null T cells, the generation memory CD8 T cells upon pathogen infection was defective [196]. Interestingly, the tumor necrosis factor (TNF) receptor-associated factor 6 (TRAF6) was also important for the FAO activation in memory cells. Mice lacking TRAF6 were unable to increase FAO and displayed defects in generation of memory cells [197]. T cell activation and proliferation may be also promoted by ROS. However, if the ROS levels become too great, some amount of T cells could undergo apoptosis, thus reducing their anticancer potential. Increased ROS production may also be provoked by proinflammatory soluble molecules (such as cytokines and chemokines) that are secreted following activation of innate immune cells and by macrophages and neutrophils, which can also produce RNS. In turn, ROS and RNS provoke serious damages at the mtDNA, causing dysfunction in the production and assembly of components of the mETC, thereby enhancing the ROS production in a dangerous loop reaction. Accordingly, loss of function of mETC is frequently associated with several cancer types, such as breast cancer [198], renal cell carcinoma [199], and thyroid carcinoma [200]. Furthermore, ROS and RNS may induce mutations of genes, and relative signaling pathways, involved in both tumor activation (oncogene) and suppression (tumor suppressor). Therefore, inflammation may lead to mutagenesis. Consistently, mutations in P53, GTPase Kras (KRAS), adenomatosis polyposis coli (APC), and wingless-related integration site (WNT) have been found in intestinal cancers characterized by chronic inflammation [201]. The inflammatory environment is also responsible to drive cell survival. Inflammatory mediators released by immune cells can converge on prosurvival pathways and increase cell proliferation and resistance to cell death. In this context, IL-1α and IL-1β promote IL-17A response, activate the NF-kB p65 subunit to cause colorectal cancer (CRC) initiation. Consistently, IL-17A was associated to poor diagnosis of CRC and inhibits the IL-1-dependent inflammation and prevented CRC development [202]. IL signaling is also responsible for promoting colitis-associated cancer. Here, IL-6 mediates cell survival and proliferation throughout the oncogenic transcription factor STAT3 [203]; alteration of the mitochondrial dynamics is an example. In fact, DRP1-FIS1-mediated mitochondrial fission increases the mitochondrial damage and boosts the inflammasome recruitment [204]. In addition, the hypoxic condition activates NLRP3. At demonstration, hypoxia in prostate cancer cell lines increase NLRP3 levels throughout NF-kB [205]. 

Furthermore, the same HIF-1α can be regulated by NLRP3. In this case, it has been observed that the adaptor protein ASC associates and stabilizes HIF-1α to increase cell migration and metastasis in oral squamous cell carcinoma [78]. However, other investigations demonstrate that hypoxia may be a repressor of NLRP3. Indeed, it has been demonstrated that hypoxic condition attenuates inflammation by reducing NLRP3 expression and that the isoform β of HIF is necessary to control the expression of NLRP3 [206]. Interestingly, in this investigation, it has been demonstrated that the activation of NLRP3 required carnitine palmitoyltransferase 1A (CPT1A)-mediated enhancement of FAO [206]. Notably, CPT1A determines the entry of long-chain fatty acids into mitochondria, leading to FAO and excessive ROS production [207]. In addition, the function of NLRP3 in cancer is controversial: some evidence highlights a protective anti-tumorigenic role; others suggest pro-tumorigenic effects [208]. For example, in breast cancer, NLRP3 increases the tumor growth and metastasis by creating an inflammatory microenvironment [209,210]. In lung cancer, NLRP3 activates the prosurvival factors Akt, the extracellular signal-regulated kinase 1/(ERK1/2), and cAMP response element-binding protein (CREB) to increase cell migration, proliferation, and invasion of cancerous cells [211]. In melanoma, NLRP3 is constitutively activated [212], and its inhibition suppresses metastasis [213]. On the contrary, NLRP3 in colorectal cancer (CRC) inhibits tumor growth, senses tissue damage, and activates cell death mechanisms against the tumor cells [214,215]. In line with this, the absence of NLRP3 was correlated with the progression of hepatocellular carcinoma [216] (Figure 3). Besides, enhanced lipid peroxidation during hypoxia-mediated inflammation contributes to the stimulation of programmed cell death (apoptosis or necroptosis) in cancer cells, resulting in cellular injury and progression of multiple pathogenesis [217,218]. Notably, it has been disclosed that tumor cells have ample lipid contents [219]. As an example, in hepatocellular carcinoma (HCC), HIF-1α augments lipid stabilization by alleviating FAO [220]. Lipid metabolism is abundant in tumor cells and exhibits a critical effect in tumorigenesis, invasion, and metastasis. To date, studies report the significant involvement of hypoxia in modulating various aspects of lipid metabolism, which are crucial for the enhanced proliferation rate, and subsequently, cancer progression [221]. The role of HIF-1 in lipid metabolism attracted more attention recently. Indeed, inhibiting HIF-1 or its target genes involved in lipid stabilization lead to significant alleviation of proliferation as well as chemoresistance in multiple cancer disorders in response to hypoxic circumstances [218,222,223,224,225]. By contrast, the overexpression of HIF-1-dependent lipid metabolism target genes is associated with cancer malignancy [218,222].

### 4.6. Hypoxia-Mediated Signaling Pathways and Cell Death in Cancer

It is well known that most cancer cells are driven toward hypoxia-directed apoptosis. Hypoxia-induced HIF-1α is stated to be associated with cancer cell malignancy and chemoradiation therapy resistance [226,227]. It is well established that HIF-1α stimulates and protects cancer cells against apoptosis. For instance, high levels of HIF-1α stabilization in pancreatic cells under hypoxia elicit their ability to resist against the apoptotic event compared with the normoxic cells through the HIF-1α-enhanced PI3K/AKT signaling pathway [228]. Although the critical role of HIF-1 in response to hypoxia-stimulated apoptosis, its molecular process in question remains unclear. Necrosis death is also involved during hypoxia in cancer. Indeed, glioma cells undergo necrosis and display resistance against prolonged hypoxia. Interestingly, this event may block the apoptotic pathway [229]. Furthermore, emerging studies have stressed the crucial role of hypoxia in ferroptosis and, given the central role of the HIF-1α element upon hypoxia, its deficiency mitigates the ferroptosis sensitivity, as it modulates the transcriptional activity of genes responsible for iron metabolisms such as transferrin receptor 1 (TFR1) and Ferritin Heavy Chain (FTH) [230]. High levels of HIF-1α protein increase the susceptibility to ferroptosis in renal carcinoma cells [231,232]. Notably, the tumor suppressor BRCA1-associated protein 1 (BAP1) is involved in apoptosis and ferroptosis in different cancer cells [233,234]. It has been demonstrated the role of BAP1 in attenuating cancer development by provoking the ferroptosis event via solute carrier family 7-member 11 (SLC7A11) inhibition [235]. Always regarding BAP1, it has been suggested that the depletion of this tumor suppressor causes enhanced HIF-1α expression through the modulation of NF-κB in uveal melanoma [236]. Taken together, given the importance of HIF-1α blockers to modulate ferroptosis in a context-dependent manner may open new directions for drug therapy [232]. Further research on the effect of hypoxia-induced HIF-1α stabilization on BAP1 would shed light on new molecular mechanisms of various events including cell death in cancer.

## 5. Novel Mechanisms and Therapeutic Targets in CVD and Cancer Disorders

HIF-1 plays critical roles in important aspects of cancer biology to allow and promote tumor cells to grow and survive in response to hypoxic conditions, including invasion, metastasis, angiogenesis, modulation of glucose and energy metabolism, and stem cell maintenance. Therefore, the poor prognosis of cancer patients results in the actual interest in studying HIF-1α as a therapeutic target in cancer disorder. Therefore, consistent with this concept, inhibiting HIF-1 or its related protein interactions has been demonstrated to block tumor proliferation. Different studies reported that HIF-1 inhibitors mitigate breast cancer metastasis in tumors sensitization to radiotherapy and mouse orthotopic transplant models [237,238]. Furthermore, several small molecules that repress HIF-1 directly or indirectly have been tested in clinical trials for different kinds of tumors [239].

By targeting HIF-1α, it is possible to intervene against malignant gliomas. Indeed, downregulation of HIF-1α by siRNA decreases both the level of matrix metalloproteinase (MMP)-2 as well as the functions of MMP-2 and MMP-9 decreasing the mobility of glioma cell via the impaired invasion-related molecules [240]. 

Small molecules blocking the expression and the functions of HIF-1 have been found effective to reduce the growth of solid tumors such as prostate and breast cancer. An example is the RNA antagonist EZN-2968, which represses HIF-1α mRNA expression [241] or PX-478, reduces the transcription of HIF-1α, blocks the translation through a VHL- and p53-independent pathway [242].

Furthermore, both PX-478 and EZN2968 elicit dose-dependent decreases in HIF-1α levels and VEGF expression, as well as the tumor size in DU145 xenograft models, and both small-molecule inhibitors were well tolerated in clinical activities [241,242]. Other molecules identified include geldanamycin, which reduces the bond of the heat shock protein 90 (HSP90) to HIF-1α to destabilize folding and increase proteasomal destruction [243]. Several HIF-1α prolylhydroxylase inhibitors that preclude VHL from binding to HIF-1α have also been developed and are now in late-stage clinical trials in disease in which HIF signaling is beneficial. For example, Roxadustat, which leads to increased endogenous erythropoietin generation, improved absorption of iron and anemia amelioration in chronic kidney disease (CKD) [244]. 

On the other hand, and in contrast to HIF-1α effect on cancer as a stimulator for the disease’s progression, HIF-1α contributes to the cardiac protection in the majority of CVDs. This depends on the duration of hypoxia and the sustain of HIF-1α activation. Different studies documented that the HIF-1 transcription factor displays a dual role in CVDs, as a protective effect in acute exposure to hypoxia, which is rapidly accumulated and stimulates a cascade of downstream target genes transcription involved in mitochondrial metabolism regulation, angiogenesis, and cell functions, leading to cardioprotection against ischemic insults. Moreover, these mechanisms regulate cellular functions in the hypoxic field to the sparsely oxygen environment and sustain normal cellular homeostasis, which under hypoxic stress is crucial for the human body. The HIF-1 pathway has a pivotal role in repairing cardiac tissue through the angiogenesis activation in I/R [245]. Thereby, it conveys a novel target to develop innovative therapies for the treatment of ischemic diseases and reduction of reperfusion injury [6]. Consistently, recent pieces of evidence reported the contribution of HIF-1α in the therapeutic effect of certain natural compounds that attenuate myocardial I/R. One example may be found for Panax notoginseng saponins, which have protective roles against I/R via HIF-1α/BCL2/BNIP3 pathway, which in return upregulates mitochondrial autophagy [246].

Nevertheless, HIF-1 contributes to the pathogenesis of various disorders during chronic exposure to hypoxia. HIF-1 is considered a promoter of atherosclerosis development, and for this reason, it may not be regarded as the disease’s therapeutic treatment. Angiogenesis induced by HIF-1 transcription is protective in the short term, but then it forms collateral vessels that may result in terrible consequences in atherosclerosis patients. Nevertheless, this hypothesis is based on current research data, which are mostly focusing on the HIF-1 signaling pathway, particularly the HIF-1α subunits. Further studies are needed on other members of the HIF family and the HIF-1α subunits. At the physiological level, there are numerous mechanisms whose modulation by these transcription factors may have great potential for treating CVDs such as atherosclerosis [247]. Pulmonary hypertension (PH) is another CVDs related to HIF-1 modulation. The proliferation of pulmonary arterial smooth muscle cells (PASMCs) and endothelial cells (Ecs) which are activated by HIF-1 and HIF-2 transcription, respectively, contributes toward the increase in pulmonary blood pressure [248]. Strategies that target the inhibition of HIF transcription in PH patients could be an interesting new perspective in the treatment of hypertension. Notably, it has been demonstrated that the HIF-2α translation inhibitor compound 76 inhibits HIF-2α, and it is capable of relieving pulmonary artery blood pressure in different models [249].

## 6. Conclusions and Future Perspectives

The mechanism of cellular response to O_2_ deficiency is primarily regulated by the HIF-1α pathway. Biological studies of HIF-1α have improved the understanding of O_2_ homeostasis and notably gained much attention recently in many research fields. In this current review, we highlighted the effect of hypoxia on mitochondrial (dys)function and inflammation in CVDs and cancer. 

We have evidenced how HIF-1α signaling exhibits divergent effects in stimulating the disease progression or inducing protection after injury in different disease conditions. This dual role particularly happens when we consider HIF-1α in CVDs and cancer. Indeed, meanwhile, HIF-1α contributes to cardiac protection in the majority of CVDs, this hypoxic factor is highly associated to tumor progression, malignancy, and resistance to chemo-radiation therapy. Regardless of these divergent effects, significant improvements of the scientific research have permitted to propose the targeting of HIF-1α as a yielding strategy for the treatment of both CVDs and cancer. 

Nevertheless, it is important to keep in mind key aspects when we are approaching treating cancer or CVDs with innovative therapies. Cancer progression contemplates different phases, which differ one from the other for several aspects. The term CVDs embraces at least 13 different conditions, which can affect both heart and blood vessels and that display its own clinical course. Furthermore, it is also fundamental to consider the heterogeneity caused by individual differences of the patients.

Only a deeper comprehension of the hypoxia-related mechanisms happening during the different phases of the tumor and in the single CVDs will really pave the way to endow and elucidate the great potential therapeutic targets and preventative approaches based on HIF-1α-modulation. 

## Figures and Tables

**Figure 1 biology-11-00300-f001:**
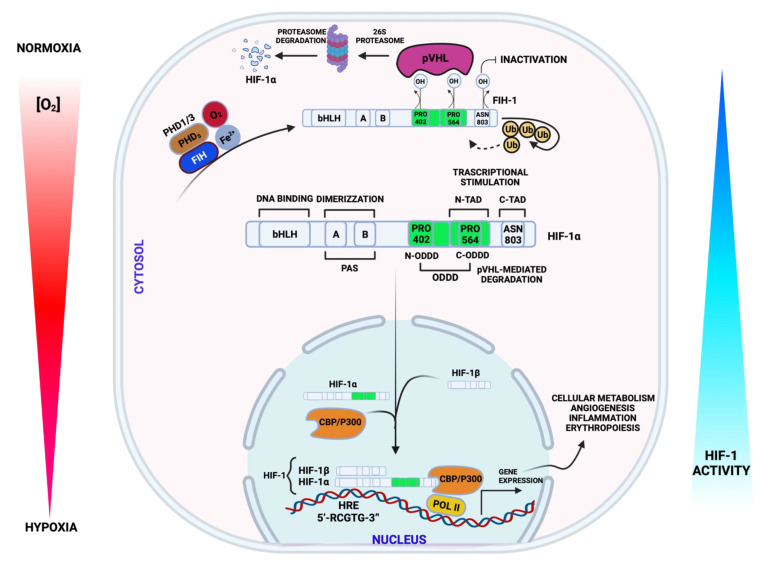
Schematic illustration representing the regulation of Hypoxic-inducible factor-1 α (HIF-1α) protein in response to normoxia and hypoxia. During normoxia, HIF-1α protein is hydroxylated by propyl-hydroxylases (PHDs) and factor inhibiting HIF (FIH). Both oxygen-dependent proteins are stimulated in normal condition and suppress HIF-1α activity. The hydroxylated prolyl residues permit the binding of HIF-1α by the von Hippel–Lindau protein (pVHL), resulting in ubiquitination and ultimate proteasomal destruction. During hypoxia or PHD inhibition, HIF-1α moves to the nucleus, heterodimerizes with HIF-1β, and subsequently binds to hypoxia response element (HRE) in the putative region of target gene to enhance their transcription.

**Figure 2 biology-11-00300-f002:**
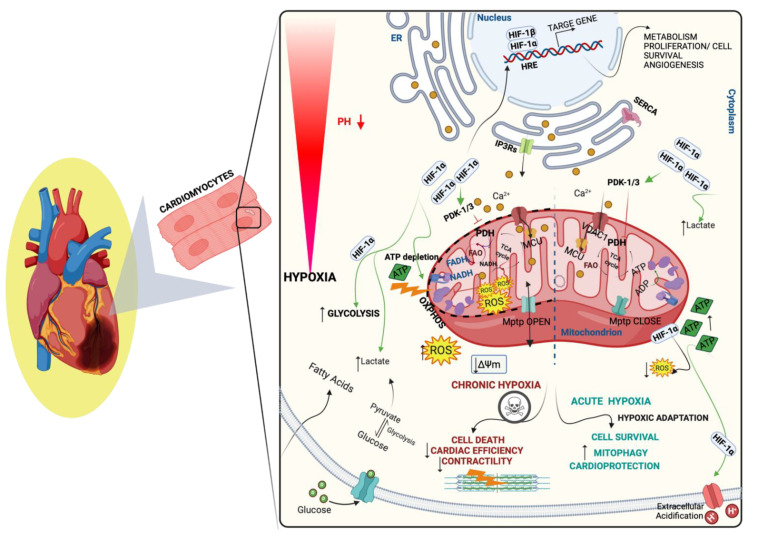
Hypoxic-inducible factor-1 α (HIF-1α) signaling and mitochondria in cardiovascular diseases (CVDs). Sudden decrease in oxygen (O_2_) levels results in abrupt biochemical and metabolic changes. Hypoxia causes the accumulation of HIF-1α that moves to the nucleus to activate genes crucial to a metabolic switch away from the mitochondrial oxidative phosphorylation system (OXPHOS) to glycolysis, the cardiomyocyte relies on anaerobic respiration instead of oxidative phosphorylation, which in turn causes disruption of the mitochondrial membrane potential (ΔΨ_m_) and adenosine triphosphate (ATP) depletion, affecting the mitochondrial Permeability Transition Pore (mPTP) opening and subsequently inhibiting contractile function. Hypoxia triggers a switch in cellular metabolism to anaerobic glycolysis, causing acidification of the cell as protons (H^+^) accumulates. Cardiomyocyte damage and mitochondrial deficiency are relatively linked to the degree of hypoxia exposure and due to the dual effect of HIF-1α; in acute hypoxia (right), HIF-1α acts as cardioprotective against oxidative damage by alleviating ROS generation and stimulating the removal of unwanted mitochondria through mitophagy. While (left) HIF-1α enhances ROS levels and increases cell death, ultimately, reduces cardiac efficiency and contractility.

**Figure 3 biology-11-00300-f003:**
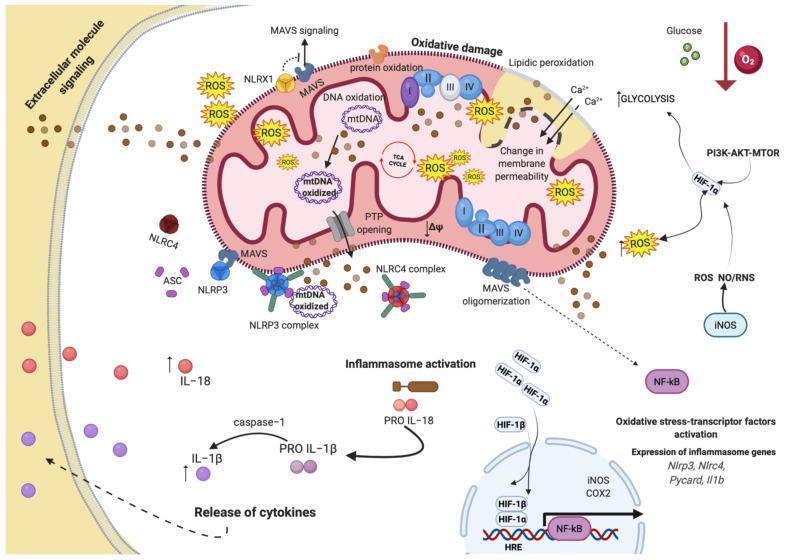
Inflammation and mitochondrial oxidative stress in response to hypoxic condition. During hypoxia, Hypoxic-inducible factor-1 α (HIF-1α) stimulates the transcription of target genes involved in inflammation and oxidative stress-transcription factors. Moreover, HIF-1α increased mitochondrial reactive oxygen species (mtROS) levels, activating nuclear factor kappa-light-chain-enhancer of activated B cells (NF-kB) transcription factor, stimulating the inflammasome genes expression, including NOD-, LRR- and pyrin domain-containing protein (NLRC)4, NLRP3, and interleukin 1β (IL1β) genes. Ultimately causing oxidative damage to the mitochondrial membrane, this event affects the membrane permeability, lipid peroxidation, and mtDNA, resulting in mitochondrial dysfunction. The FosfoInositide-3-Kinasi (PI3K)-protein kinase B (AKT)-mechanistic target of rapamycin (MTOR) pathway upregulates HIF-1α during hypoxia.

**Table 1 biology-11-00300-t001:** Representative list summarizing the effect of HIF-1α in cardiovascular diseases. HIF-1α*^tg^* mice: HIF-1α*^tg^* transgenic mice models. CH rodent models: chronic hypoxic rodent models. LPS: lipopolysaccharide.

Cardiovascular Disorders	In Vivo/In Vitro	Animal Models	Cell Lines	HIF-1α Effect	References
Ischemia-reperfusion injury (I/R injury)	In vitro	-	Rat neonatal ventricular cardiomyocytes cells	Cardioprotective effect, overexpression of HIF-1α elevated target genes (iNOS, VEGF, HSP70, and GLUT1-4)	[32]
	In Vivo/in vitro	PHD3^−/−^ mice	HL-1 cardiomyocytes	Cardioprotection, PHD3 deletion increased HIF-1α, resulted in cardiomyocytes death suppression	[30,31]
	In Vivo/in vitro	Sprague Dawley (SD)/rat model	H9C2 cardiomyoblasts	Cardioprotection, BNIP3-mediated autophagy modulation	[39]
Myocardial infraction (MI)	In Vivo	Post-MI mice	-	Cardioprotection, upregulated angiogenesis	[57]
	In vivo/in vitro	MI-mice	Rat neonatal cardiomyocytes	Detrimental, stimulated apoptosis through p53 following MI	[58]
Heart failure (HF)	In vivo	HIF-1α*^tg^*mice	-	Detrimental, prolonged HIF-1α accumulation increased disease development	[28]
Myocarditis	In vitro	-	H9C2 cardiomyoblasts	Detrimental, repression of HIF-1α improved cardiomyocytes at odds with LPS-stimulated cell death	[29]
Dilated cardiomyopathy	In vivo	PHD2^−/−^ mice	-	Detrimental, prolonged HIF-1α upregulation promoted dilated cardiomyopathy	[38]
Cyanotic congenital heart disease (CCHD)	In Vivo	CH rodent models	-	Cardioprotection, HIF-1α overexpression alleviated maladapted metabolic	[59]
Cardiac hypertrophy	In Vivo	HIF-1α KO mice	-	Cardioprotection by controlling negatively TGF-β	[60]

**Table 2 biology-11-00300-t002:** Representative list summarizing the effect of HIF-1α on mitochondria in cancer.

Cancer Disorders	In Vivo/In Vitro	Animal Models	Cell Lines	HIF-1α Effect	References
Human Breast ductal carcinoma	In vitro	-	MCF-7 cells	Inhibited ER Estrogen receptor expression	[62]
Renal carcinoma	In vitro	-	RCC4 and RCC10	Increased mitochondrial biogenesis	[69]
Hepatocellular carcinoma (HCC)	In vivo	Mice and Diethylnitrosamine model of Murine HCC	HCC cell lines	Promoted mitochondrial biogenesis and reduced ATP	[70]
Triple negative breast cancer (TNBC)	In vitro/in vivo	Nude mice	MDA-MB-231	Enhanced mitochondrial OXPHOS and elevated ROS generation	[71]
	In vitro	-	MDA-MB-231 and SUM-149 cells	Increased intracellular glutathione levels	[72]
	In vivo	MMTV-PyMT mice	Primary MECs	Regulated mitochondrial mass	[73,74]
Colorectal cancer	In vitro/in vivo	Oma1^−/−^ mice	HCT116 cells	Increased mitochondrial ROS	[75]
Several human cancers	In vitro	-	A549, CCL39, HeLa, LS174, MCF7, PC3, ORL33, SKMel, and 786-O cells	Enlarged mitochondrial phenotype	[76]
Glioblastoma	In vitro/in vivo	Foxp3-*^YFP-CRE^* × HIF-1α -*^fl/fl^* mice	Murine glioma GL-261	Promoted fatty acids oxidation for mitochondrial metabolism	[64]
Cancer cachexia (CC)	In Vitro/in vivo	C26 mice model	Colon-26 (C26) adenocarcinoma	Affected the metabolic changes	[77]
Oral cancer	In vitro	-	Oral squamous cell carcinoma (OSCC)	Stimulated migration and invasion in the indicated cells	[78]

## Data Availability

Not applicable.

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
