# Peer review of "The Interplay of Hypoxia Signaling on Mitochondrial Dysfunction and Inflammation in Cardiovascular Diseases and Cancer: From Molecular Mechanisms to Therapeutic Approaches"

_biology, 2022, doi:10.3390/biology11020300_

Round 1
Reviewer 1 Report
Authors have reviewed hypoxia mediating signaling pathways on mitochondrial function, and inflammation in key human diseases such as, CVDs and cancer. The paper is written well; however, following needs to be addressed:
- Why did authors select two diseases i.e. CVD and cancer? HIF plays role in many other diseases like diabetes etc. Authors need to give clear rationale behind two disease selection in introduction section.
- Authors did mention about Warburg effect; however, they have entirely missed to address cancer associated cachexia in which hypoxia has major role to play.
- A section on mitophagy is also needed to be discussed since there is a major interplay between hypoxia-mitochondrial functions and mitophagy. Authors have briefly covered autophagy in few lines, but addition of mitophagy will increase interest of the readers.
- Authors should put 1 or 2 tables summarizing studies of HIF in different cell lines/animal models for quick glance.
Author Response
Reviewer 1
Comments and Suggestions for Authors
Authors have reviewed hypoxia mediating signaling pathways on mitochondrial function, and inflammation in key human diseases such as, CVDs and cancer. The paper is written well; however, following needs to be addressed:
- Why did authors select two diseases i.e. CVD and cancer? HIF plays role in many other diseases like diabetes etc. Authors need to give clear rationale behind two disease selection in introduction section.
We thank the reviewer for the insightful comment, we have expanded our choice in selecting both disorders accordingly.
- Authors did mention about Warburg effect; however, they have entirely missed to address cancer associated cachexia in which hypoxia has major role to play.
We thank the reviewer for the important suggestion. We have included cancer cachexia as recommended.
- A section on mitophagy is also needed to be discussed since there is a major interplay between hypoxia-mitochondrial functions and mitophagy. Authors have briefly covered autophagy in few lines, but addition of mitophagy will increase interest of the readers.
Thank you for the valuable comment we have included hypoxia-mediated mitophagy in both mentioned disorders.
- Authors should put 1 or 2 tables summarizing studies of HIF in different cell lines/animal models for quick glance.
We thank the reviewer for the comment, we have been able to incorporate the tables as highlighted.
Reviewer 2 Report
This review article summarized the current comprehensions about the effect of hypoxia-mediating signaling pathways on mitochondrial function, and inflammation in CVDs and cancer. Moreover, potential therapies involved in hypoxia and mitochondrial dysfunction in these conditions also discussed. Based on this perspective, this review from Bouhamida and colleagues is very timely and has the potential to provide valuable information on this complex process. However, I have some specific concerns that should be addressed.
Major concerns:
Cell death plays an important role in the regulation of inflammation and may be the result of inflammation. I recommend to the authors adding the description about the effects of hypoxia-mediating signaling pathways on programmed cell death process. Specifically, it would be more informative if the authors could illustrate the roles of hypoxia-mediating signaling pathways on programmed cell death (apoptosis, ferroptosis, necroptosis, etc.) of cardiomyocytes and cancer cells.
Minor concerns:
- Figure 1, PHD1/3 should be moved to the right close to the arrow.
- Figure 1, in the nucleus, HIF-1β should be moved to the top and HIF-1α should be moved to the bottom.
- Page 5, line 170, “HF” should be “heart failure (HF)” and please check if reference 33 is correct or not?
Author Response
Reviewer 2
Comments and Suggestions for Authors
This review article summarized the current comprehensions about the effect of hypoxia-mediating signaling pathways on mitochondrial function, and inflammation in CVDs and cancer. Moreover, potential therapies involved in hypoxia and mitochondrial dysfunction in these conditions also discussed. Based on this perspective, this review from Bouhamida and colleagues is very timely and has the potential to provide valuable information on this complex process. However, I have some specific concerns that should be addressed.
Major concerns:
Cell death plays an important role in the regulation of inflammation and may be the result of inflammation. I recommend to the authors adding the description about the effects of hypoxia-mediating signaling pathways on programmed cell death process. Specifically, it would be more informative if the authors could illustrate the roles of hypoxia-mediating signaling pathways on programmed cell death (apoptosis, ferroptosis, necroptosis, etc.) of cardiomyocytes and cancer cells.
We appreciate this recommendation; we have added a section on hypoxia-mediated programmed cell death in both CVDs and cancer.
Minor concerns:
- Figure 1, PHD1/3 should be moved to the right close to the arrow.
We thank the reviewer for the comment, we have accordingly modified figure 1.
- Figure 1, in the nucleus, HIF-1β should be moved to the top and HIF-1α should be moved to the bottom.
Agree; we thank the reviewer for pointing this out, and we have also modified the other figures as well.
- Page 5, line 170, “HF” should be “heart failure (HF)” and please check if reference 33 is correct or not?
We thank the reviewer for the observation, we have incorporated a new reference throughout the manuscript.
Reviewer 3 Report
Manuscript ID # 1560797
In this review title “The interplay of hypoxia signaling on mitochondrial dysfunction, and inflammation in cardiovascular diseases and cancer: from molecular mechanisms to therapeutic approaches” authors have reviewed the role of HIF transcription factors in the pathophysiology of cardiovascular diseases (CVD) and cancer via regulating mitochondria derived ROS and integrated inflammation. HIF transcription factors (specially HIF-1α) is the central player in the regulation of different signaling pathways and gene expression under hypoxia condition which is an important area of research. The authors have described the context-dependent role of HIF-1α in the regulation of CVD and cancer pathology. I have the following comments listed below:
Comments:
- The authors have written the manuscript very well with listed references.
- Although authors have described the role of HIF-1α in the regulation of mitochondrial ROS and inflammation the pathophysiology of CVD and cancer. However, it has a context-dependent effect whereas in cardiovascular disease (majority) it has a protective effect whereas in cancer it promotes the cell proliferation and development of cancer.
- Lipid metabolisms play a significant role in CVD and cancer progression. Although, authors have discussed the role of mtROS and inflammation in the development of disease pathology, however, they have not discussed the contribution of HIF-1α in lipid metabolism in response to mtROS and inflammation. Authors should include these points in the current manuscript.
- Authors have stated that HIF-1α can be a target for the development of therapeutics against disease conditions. Since HIF-1α has a context-dependent effect, means in some cases it has a protective effect (CVD) but in other cases, it promotes the pathology (cancer). Therefore, HIF-1α cannot be a good target for the development of common therapeutics. Authors should explain this with focused rationale in particular disease conditions and not the common effect.
- In this review, the authors have explained the contribution of HIF-1α in the regulation of mt ROS and inflammation in CVD and cancer based on the literature available. They have put the information whatever it was available. However, they have not given the direction of future studies based on the HIF-1α target. They should discuss whatever they are thinking about HIF-1α targeted development of disease.
- Authors should discuss why HIF-1α is important as a therapeutic target and why they have chosen it as a target.
Author Response
Reviewer 3
Comments and Suggestions for Authors
In this review title “The interplay of hypoxia signaling on mitochondrial dysfunction, and inflammation in cardiovascular diseases and cancer: from molecular mechanisms to therapeutic approaches” authors have reviewed the role of HIF transcription factors in the pathophysiology of cardiovascular diseases (CVD) and cancer via regulating mitochondria derived ROS and integrated inflammation. HIF transcription factors (specially HIF-1α) is the central player in the regulation of different signaling pathways and gene expression under hypoxia condition which is an important area of research. The authors have described the context-dependent role of HIF-1α in the regulation of CVD and cancer pathology. I have the following comments listed below:
Comments:
- The authors have written the manuscript very well with listed references.
We thank you for your honest comment that we appreciate.
- Although authors have described the role of HIF-1α in the regulation of mitochondrial ROS and inflammation the pathophysiology of CVD and cancer. However, it has a context-dependent effect whereas in cardiovascular disease (majority) it has a protective effect whereas in cancer it promotes the cell proliferation and development of cancer.
We thank the reviewer. We have addressed this point as requested
- Lipid metabolisms play a significant role in CVD and cancer progression. Although, authors have discussed the role of mtROS and inflammation in the development of disease pathology, however, they have not discussed the contribution of HIF-1α in lipid metabolism in response to mtROS and inflammation. Authors should include these points in the current manuscript.
We thank the reviewer for the recommendation, we have included these points accordingly
- Authors have stated that HIF-1α can be a target for the development of therapeutics against disease conditions. Since HIF-1α has a context-dependent effect, means in some cases it has a protective effect (CVD) but in other cases, it promotes the pathology (cancer). Therefore, HIF-1α cannot be a good target for the development of common therapeutics. Authors should explain this with focused rationale in particular disease conditions and not the common effect.
Thank you for pointing this out, we revised this part as recommended.
- In this review, the authors have explained the contribution of HIF-1α in the regulation of mt ROS and inflammation in CVD and cancer based on the literature available. They have put the information whatever it was available. However, they have not given the direction of future studies based on the HIF-1α target. They should discuss whatever they are thinking about HIF-1α targeted development of disease.
We thank the reviewer, we have done further modifications.
- Authors should discuss why HIF-1α is important as a therapeutic target and why they have chosen it as a target.
We agree with this, and we have incorporated this point accordingly.
In addition to the above comments, all spelling and grammatical errors pointed out by the reviewers have been corrected.
Round 2
Reviewer 1 Report
Authors have incorporated all the suggestion.
Reviewer 2 Report
The authors have addressed all of my comments and revised the manuscript accordingly.